# Successful Treatment of Brain Abscess Caused by *Nocardia farcinica* with Combination Therapy Despite Discrepancies in In Vitro Results: A Case Report and Review of Diagnostic and Therapeutic Challenges

**DOI:** 10.3390/microorganisms13071536

**Published:** 2025-06-30

**Authors:** Eva Larrañaga Lapique, Salomé Gallemaers, Sophie Schuind, Chiara Mabiglia, Nicolas Yin, Delphine Martiny, Maya Hites

**Affiliations:** 1Departement Clinic of Infectious Diseases, Hôpital Universitaire de Bruxelles (HUB)-Erasme, 1070 Brussels, Belgium; 2Department of Microbiology, Laboratoire Hospitalier Universitaire de Bruxelles-Universitaire Laboratorium Brussel (LHUB-ULB), Université Libre de Bruxelles (ULB), 1000 Brussels, Belgium; salome.gallemaers@lhub-ulb.be (S.G.);; 3Department of Neurosurgery, Hôpital Universitaire de Bruxelles (HUB)-Erasme, 1070 Brussels, Belgium; 4Department of Radiology, Hôpital Universitaire de Bruxelles (HUB)-Erasme, 1070 Brussels, Belgium; 5Faculty of Medicine and Pharmacy, University of Mons (UMONS), 7000 Mons, Belgium

**Keywords:** *Nocardia farcinica*, brain abscess, antimicrobial susceptibility testing (AST), management

## Abstract

*Nocardia* spp. is an environmental Gram-positive bacterium able to cause infections in humans, predominantly of an opportunistic nature. Nocardial brain abscesses are rare and result from dissemination from another primary lesion, mainly observed in immunocompromised hosts. The diagnosis of nocardiosis relies on direct examination and bacterial culture, but antimicrobial susceptibility testing (AST) remains controversial due to technical challenges, limited standardization, and a paucity of studies correlating in vitro susceptibility with clinical efficacy. Management is challenging and usually based on expert opinion, as robust evidence is limited. In this case report, we describe an immunocompromised patient with a *Nocardia farcinica* brain abscess who achieved clinical resolution following combination therapy that included ceftriaxone, despite in vitro resistance, illustrating the complexities in interpreting AST and guiding treatment decisions in rare infections.

## 1. Introduction

*Nocardia* spp. are ubiquitous environmental Gram-positive bacilli that primarily cause opportunistic infections in humans [1], particularly in patients with impaired cell-mediated immunity [2]. Since its first description by Edmond Nocard in 1988, more than 100 species have been reported among the genus *Nocardia* spp.; the most frequently reported species from clinical sources are *Nocardia farcinica*, *N. nova*, *N. cyriacigeorgica*, *N. brasiliensis,* and *N. abscessus* [3].

Pulmonary nocardiosis is the most frequent site of infection, reflecting its acquisition via inhalation [1]. Dissemination to other organs such as the brain, skin, soft tissues, and/or other sites is common, particularly in the case of immune deficiency [2,4]. Isolated *Nocardia* infection of the central nervous system (CNS) is rare, occurring almost exclusively in immunocompromised individuals, and is associated with high mortality rates [5]. In case of CNS involvement, cerebral abscesses are the most frequently reported presentation [6]. They may be asymptomatic, therefore highlighting the need to systematically perform brain imaging in all patients with suspicion of nocardiosis [1,7].

The diagnosis of nocardia infection relies on both direct examination and bacterial culture [8]. The subsequent antibiotic therapeutic management of the infection depends on the challenging interpretation of antimicrobial susceptibility testing (AST).

*N*. *farcinica* is one of the most frequently encountered *Nocardia* subspecies in human infection. It is associated with disseminated disease, poorer clinical outcomes, a high tropism for CNS involvement, and, notably, the highest level of multi-antibiotic resistance among all *Nocardia* subspecies [2,5]. Furthermore, in vitro resistance to third-generation cephalosporins is frequently reported [9].

As with other rare disease entities, treatment of CNS nocardiosis is based on expert opinion and retrospective reviews. Current guidelines recommend that antimicrobial therapy should be started as soon as the diagnosis of nocardiosis is made; treatment should reach all infected sites, be active against the *Nocardia* spp. concerned and take into consideration the patient’s comorbidities and concomitant medications. Concomitantly to appropriate and adequate antibiotic therapy, reduction, when possible, of immunosuppressive therapy should also be performed [1,6,10], and a neurosurgical approach should be considered [6,10,11,12].

This review will discuss the microbiological diagnosis, the challenges of AST, and the therapeutic options for patients with *Nocardia farcinica* brain abscesses through the description of the case of an immunocompromised patient successfully treated with combination therapy, including ceftriaxone, despite AST showing resistance.

## 2. Case Report

A 30-year-old afebrile woman was admitted with a headache and mild aphasia of subacute onset. Her medical history was remarkable for refractory severe myasthenia gravis for which she was receiving methylprednisolone (40 mg/day) and azathioprine (75 mg/day). Physical examination revealed only minor right upper limb paresis. Magnetic resonance imaging (MRI) of the brain showed a large polylobed left frontal brain abscess with significant perilesional edema (Figure 1(1A,1B)). Neurosurgical drainage by small craniotomy was performed on day 1, without complications. Direct examination of the aspirated fluid revealed numerous neutrophils and long, filamentous, branching Gram-positive bacilli. The specimen was cultured on standard media, and after several days of incubation, aerobic growth of chalky white colonies was observed (Figure 2). MALDI-TOF mass spectrometry initially identified the isolate as *Nocardia* spp., but the top ten taxonomic matches included various species, such as *Nocardia kroppenstedtii*, *Nocardia asteroides,* leading to uncertainty regarding the exact species. Due to this ambiguity and the clinical relevance of precise identification, 16S rRNA gene sequencing was performed, which confirmed the species as *Nocardia farcinica*. No primary lesion was identified on either the skin or in the lungs, despite thorough dermatological examination and a chest computed tomography scan.

Table 1 summarizes the treatment received by our patient. Empirical therapy after the surgery included high-dose ceftriaxone (2 g twice daily). On day 4, trimethoprim/sulfamethoxazole (TMP/SMX—10 mg/kg/day of the trimethoprim component) was added to the initial regimen when *Nocardia* spp. was identified by MALDI-TOF. Given the limitations of MALDI-TOF in providing definitive species-level identification and the clinical importance of accurate speciation, 16S rRNA sequencing was performed, confirming *Nocardia farcinica* as the causative pathogen. Ceftriaxone was subsequently replaced with imipenem (500 mg three times daily) on day 8. Immunosuppression was reduced by stopping azathioprine, while corticosteroids were maintained for cerebral edema; no recurrence of myasthenia gravis was observed. On day 14, an IgE-mediated hypersensitivity type 1 reaction (rash) to imipenem was observed, prompting a therapeutic change from imipenem to high-dose ceftriaxone, while pursuing TMP-SMX.

The initial AST was conducted in our laboratory using gradient diffusion strips (Etest^®^, bioMérieux, Marcy-l’Étoile, France) on Mueller–Hinton agar plates (bioMérieux, Marcy-l’Étoile, France). This analysis was performed twice, both times yielding consistent results. The readings were taken after 24, 48, and 72 h of incubation. The *Nocardia* strain was found to be susceptible to ceftriaxone (MIC 2 µg/mL), imipenem (MIC 0.25 µg/mL), cotrimoxazole (MIC 0.125 µg/mL), amoxicillin-clavulanate, linezolid, amikacin (MIC 2 µg/mL), and moxifloxacin. To corroborate the AST results, the original strain was sent to the French National Reference Center (NRC) for *Nocardia*. The reference method of broth microdilution was used; it revealed resistance to ceftriaxone (MIC 64 µg/mL). The complete AST results obtained with both methods are presented in Table 2.

The dual initial antibiotic treatment of TMP/SMX and high-dose ceftriaxone was continued for a period of 12 weeks, resulting in a favorable clinical and radiological response. Maintenance therapy was continued with TMP/SMX alone. Unfortunately type II hypersensitivity (hemolytic anemia) and digestive intolerance (nausea and vomiting) developed on day 210, followed by complete resolution after discontinuation of TMP/SMX. The therapeutic plan based on international recommendations [6,10,13] was to administer at least 12 months of antibiotic treatment before beginning secondary prophylaxis. Choosing the agent to pursue maintenance therapy was a challenge as quinolones were contraindicated (e.g., myasthenia), long-term linezolid therapy was foreseen to be difficult to manage because of potential side effects, and the patient refused to take tetracyclines (fear of mucocutaneous mycosis and phototoxicity). By this time, the patient was performing well, and the brain lesion showed significant improvement on MRI (Figure 1(2A,2B)); therefore, the decision was taken to pursue maintenance therapy with ceftriaxone in the absence of other suitable alternatives. This antibiotic was well tolerated by the patient in the past, and the pathogen showed susceptibility to ceftriaxone based on the Hodge gradient strip method. After 15 months of treatment (day 488), maintenance therapy was stopped because clinical and radiological responses were maintained. Secondary prophylaxis was initiated and pursued with tetracyclines (doxycycline 200 mg per day), after extensively discussing the pros and cons of this treatment with the patient. The patient is currently performing well at more than 24 months of follow-up.

## 3. Discussion

We describe the case of a *N. farcinica* cerebral abscess in an immunocompromised patient in the context of AST discrepancies, highlighting the complexity of diagnosis and therapeutic aspects of Nocardia CNS infections.

### 3.1. Microbiological Diagnosis

Our case illustrates several microbiological challenges when managing *Nocardia* spp. infections in humans. Traditional biochemical methods are insufficient for accurately differentiating *Nocardia* species, especially given the growing number of *Nocardia* species identified in human infections over time. MALDI-TOF mass spectrometry is commonly used for genus-level identification but has limitations at the species level, as the large number of existing *Nocardia* species and their rarity contribute to incomplete databases. Only the most common species, such as *N. brasiliensis*, *N. cyriacigeorgica*, *N. farcinica*, *N. nova/N. nova complex*, and *N. otitidiscaviarum*, are consistently identified when the identification score is sufficient [14]. The gold standard for species-level identification is polymerase chain reaction (PCR) sequencing, specifically targeting the 16S rRNA gene. This method provides reliable species-level identification in most cases [10], as was the case for our patient. Species identification is crucial because different *Nocardia* species display variable resistance profiles, directly impacting the choice of antimicrobial therapy and patient outcomes [14,15].

Performing and interpreting AST for *Nocardia* species also presents several challenges. Different AST techniques, such as broth microdilution, E-tests, or disk diffusion, may be used; however, these methods can yield inconsistent results. Broth microdilution, although standardized by the CLSI for *Nocardia* [15] and considered to be the reference method, has significant limitations. The CLSI has cautioned against potential false-resistant results, particularly for ceftriaxone against *N. brasiliensis*, and for imipenem against *N. farcinica* [15]. Additionally, broth microdilution is a time-consuming method that requires skilled personnel; therefore, it is rarely performed outside CNRs. As a result, most laboratories opt for simpler methods like disk diffusion or E-tests, which are quicker and yield AST results generally comparable for most antibiotics [16,17,18]. Nevertheless, discrepancies may still occur with certain drugs, particularly ceftriaxone, imipenem, tigecycline, and sulfonamides [8,9,10], as observed in our patient, concerning *Nocardia farcinica* susceptibility to ceftriaxone. An explanation for the conflicting ceftriaxone AST results observed in our patient could be related to the production of inducible β-lactamase by *Nocardia* spp. The expression of this enzyme can vary depending on the experimental conditions, leading to discrepancies in β-lactam susceptibility results when using different AST methods [8].

Despite that AST is considered essential for clinical management of patients with *Nocardia* infections, data correlating in vitro activity with clinical efficacy for this infection is lacking [10], thus making it difficult to fully understand the clinical implications of AST results. As a result, any isolate that does not display the expected antimicrobial susceptibility pattern for its species should be retested or sent to a reference laboratory for result confirmation [8,9,18].

### 3.2. Treatment of CNS Nocardiosis

Regarding therapeutic options in *Nocardia* infections, sulfonamides, most commonly trimethoprim-sulfamethoxazole (TMP-SMX), are the agents of choice because of their reliable in vitro activity against *Nocardia* spp. [19], the high drug concentrations achieved in infected tissues, and their status as a cornerstone of nocardiosis treatment for years, with few reported cases of clinical failure [10]. Other antimicrobials that may have good activity against *Nocardia* spp. are carbapenems (imipenem and meropenem), parenteral third-generation cephalosporins (ceftriaxone and cefotaxime), tetracyclines (minocycline), amikacin, quinolones, linezolid, or ampicillin and ampicillin-clavulanate [9,10,16,17,18]. Unfortunately, no randomized clinical trials have yet been performed to compare the efficacy of different antibiotic regimens or mono versus combination therapy for nocardiosis. Treatment is usually divided into an initial therapy phase (before species identification and AST results), followed by a maintenance therapy phase. Recommended initial treatment in patients with non-severe nocardiosis is TMP/SMX alone, while combination regimens, usually with third-generation cephalosporins, imipenem, or amikacin, should be administered if patients present a severe, disseminated, and/or CNS infection [10,13]. An alternative is the combination of linezolid with imipenem [10]. Combination therapy should be continued until clinical improvement occurs and *Nocardia* spp. identification with AST n is obtained [10]. Most patients respond to initial therapy and can be shifted to maintenance therapy with a single oral regimen to complete treatment. The optimal duration of therapy is unknown, but treatment courses of 4–6 months are typically recommended for pulmonary and soft tissue infections, while duration is usually extended to 12 months or longer in patients who have disseminated or CNS disease, to prevent recurrence [10]. A neurosurgical approach should be considered to allow for microbiological documentation (particularly in the absence of other primary lesions), exclude co-infections, and achieve source control [6,10,11,12] when there is CNS involvement. The optimal strategy between needle aspiration and surgical excision is still a matter of debate, but should take into consideration the patient’s clinical status, the size of the abscess, and the degree of brain swelling and compression [12].

We decided to continue ceftriaxone monotherapy during the maintenance phase despite discordant results between AST methods (gradient strip method and broth microdilution), given the favorable evolution of the patient with ceftriaxone treatment (from Day 14 to Day 98), in the absence of a suitable alternative (TMP-SMX in particular), and the need to pursue maintenance therapy for at least 12 months. It is advised that therapeutic modifications for *Nocardia* spp. infections should consider response to initial therapy, AST results, and individual specificities [6,10]. In this case, despite the contradictory laboratory results, the patient’s favorable clinical response played a decisive role in guiding treatment. The positive clinical outcome suggests that the in vitro resistance reported by the NRC may not have translated into a clinical lack of efficacy.

We acknowledge that our case report has limitations: in addition to describing a single clinical case, our patient initially received combination antibiotic therapy, making it difficult to attribute the favorable outcome solely to ceftriaxone monotherapy. Nevertheless, we do not believe that the favorable clinical and radiological response to treatment could be exclusively attributed to the patients’ six-month treatment with TMP-SMX either, as this would be inconsistent with the recent literature recommending at least 9–12 months of antibiotic therapy for SNC nocardiosis [10,11,13]. This case reflects the diagnostic and therapeutic dilemmas clinicians face in real-life practice and highlights the gap between laboratory data and real-world outcomes.

## 4. Conclusions

Antimicrobial susceptibility testing for Nocardia species is a valuable tool that provides comparative data and may be particularly useful in the management of resistant strains, such as *N. farcinica*, as a guide to therapy. Nevertheless, *Nocardia* spp. AST interpretation should be approached with caution, as it has not been subject to rigorous correlations with clinical outcomes. Consequently, in rare and challenging infections such as CNS nocardiosis or in cases where AST results for *Nocardia* spp. infections are inconclusive or conflicting, therapeutic decisions should be individualized and guided by clinical judgment, rather than relying solely on AST results.

## Figures and Tables

**Figure 1 microorganisms-13-01536-f001:**
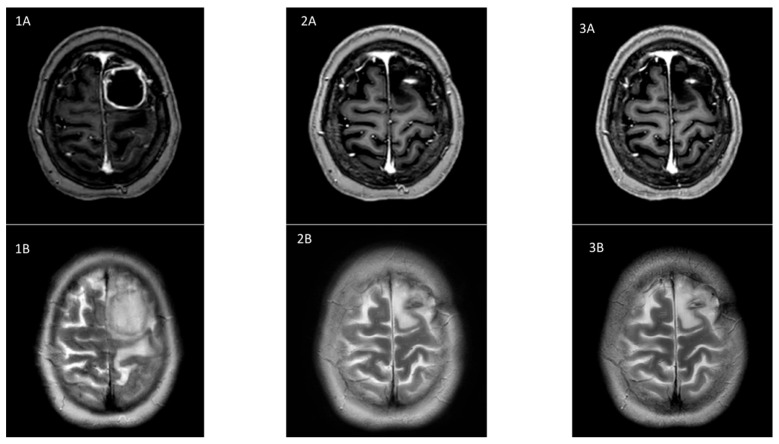
Evolution of cerebral MRI over time. (**1A**): T1 image after Gadolinium injection shows a polylobed left frontal brain abscess (48 × 45 mm) with peripheral enhancement on admission before any treatment (surgical or antimicrobial). (**1B**) T2 image shows a significant perilesional vasogenic edema at the same moment. (**2A**,**2B**): show MRI after surgical excision and 6 months of antibiotic treatment with regression of the lesion and edema. (**3A**,**3B**) show stability of the sequelae at 18 months follow-up.

**Figure 2 microorganisms-13-01536-f002:**
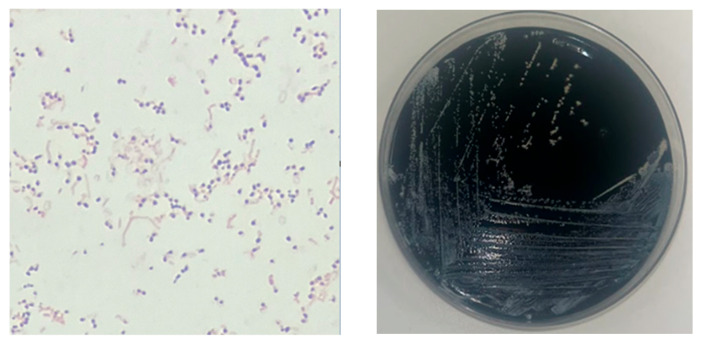
Direct examination at ×100 magnification showing long, filamentous and branching Gram-positive bacilli (**left**); chalky-white colonies of *Nocardia* spp. on culture medium after 48 h of incubation (**right**).

**Table 1 microorganisms-13-01536-t001:** Timeline of antibiotic therapy received by the patient.

Initial therapy	
Day 1	- High-dose ceftriaxone empirically started
Day 4	- Addition of IV trimethoprim-sulfamethoxazole (TMP/SMX) after *Nocardia* spp. identification
Day 8	- Replacement of ceftriaxone with imipenem after N. species *farcinica* identification - IV TMP/SMX continued
Day 14	- Switch imipenem to high-dose ceftriaxone because of type I hypersensitivity to imipenem - IV TMP/SMX continued
Day 28	- Shift IV TMP/SMX to oral - High-dose ceftriaxone pursued via maintained in OPAT (outpatient parenteral antimicrobial therapy) program
**Maintenance therapy**	
Day 98	- Stop ceftriaxone - Maintenance phase with TMP/SMX alone
Day 210	- Discontinuation of TMP/SXT because of type II hypersensitivity and digestive intolerance
Day 211	- New initiation of high-dose ceftriaxone in monotherapy as maintenance phase despite discordant results between AST methods based on the need to pursue therapy for at least one year, excellent tolerance, absence of suitable alternatives, and good clinical and radiological responses
**Secondary prophylaxis**	
Day 488	- End of maintenance therapy with ceftriaxone - Doxycycline started for secondary prophylaxis

The initial therapy is in yellow, the maintenance phase in green, and the secondary prophylaxis in blue.

**Table 2 microorganisms-13-01536-t002:** Comparison of antimicrobial susceptibility results for a *Nocardia* spp. isolate using gradient strip method (LHUB-ULB) and reference broth microdilution (CNR). Interpretation according to CLSI M62 guidelines. MIC: Minimal Inhibitory Concentration; S: Susceptible; R: Resistant; I: Intermediate.

Antibiotic	Gradient Strip Method (ULB-LHUB)		Broth Microdilution (French CNR)	
	MIC (μg/mL)	Interpretation	MIC (μg/mL)	Interpretation
Amoxicillin-Clavulanate	2	S	8	S
Ceftriaxone	8	S	64	R
Imipenem	0.25	S	4	S
TMP/SMX	0.125	S	1	S
Amikacin	2	S	1	S
Moxifloxacin	<0.25	S	<0.25	S
Minocycline	-	-	2	I

## Data Availability

The original contributions presented in this study are included in the article. Further inquiries can be directed to the corresponding author.

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
