# Peer review of "Successful Treatment of Brain Abscess Caused by Nocardia farcinica with Combination Therapy Despite Discrepancies in In Vitro Results: A Case Report and Review of Diagnostic and Therapeutic Challenges"

_microorganisms, 2025, doi:10.3390/microorganisms13071536_

Round 1
Reviewer 1 Report
Comments and Suggestions for Authors
This is a well-documented and engaging case report describing the successful management of a brain abscess caused by Nocardia farcinica with ceftriaxone, despite its in vitro resistance. The manuscript provides valuable clinical insights into the limitations of antimicrobial susceptibility testing (AST) and the need to balance laboratory data with clinical judgment, particularly in rare and challenging infections such as CNS nocardia.
The authors successfully integrate a detailed patient case with an extensive discussion of diagnostic limitations, therapeutic uncertainties, and realistic management strategies. This paper is timely, clinically relevant, and fully consistent with the scope of the journal.
- Although the variability in AST results is the main theme, the timeline and rationale for maintaining ceftriaxone despite the National Research Council findings could be further explained in the Discussion section for greater clarity.
- The description of Figure 1 (MRI Evolution) could be improved by adding more annotations to each sub-image (1A, 1B, 2A, etc.) to help the reader visually follow the progression.
- There are some minor grammatical errors (e.g., the phrase "ineffective" in line 184 could be replaced with "lack of efficiency"). A simple proofreading by a native English speaker is recommended.
- A concise table summarizing the antimicrobial susceptibility profile (initial E-test versus NRC method) and treatment schedule may help clarify the picture for the reader.
Comments on the Quality of English Language
There are some minor grammatical errors (e.g., the phrase "ineffective" in line 184 could be replaced with "lack of efficiency"). A simple proofreading by a native English speaker is recommended.
Author Response
|
Response to Reviewer 1 Comments
|
||
|
1. Summary |
|
|
|
Thank you very much for taking the time to review this manuscript. Please find the detailed responses below and the corresponding corrections highlighted in the re-submitted files.
|
||
|
2. Questions for General Evaluation |
Reviewer’s Evaluation |
Response and Revisions |
|
Does the introduction provide sufficient background and include all relevant references? |
Yes |
|
|
Are all the cited references relevant to the research? |
Yes |
|
|
Is the research design appropriate? |
Yes |
|
|
Are the methods adequately described? |
Yes |
|
|
Are the results clearly presented? |
Can be improved |
We have revised presentation + added a new figure + a new table |
|
Are the conclusions supported by the results?
|
Yes |
|
|
3. Point-by-point response to Comments and Suggestions for Authors
|
||
|
Comments 1: Although the variability in AST results is the main theme, the timeline and rationale for maintaining ceftriaxone despite the National Research Council findings could be further explained in the Discussion section for greater clarity. |
||
|
Response 1: Thank you for pointing this out. We agree with this comment. Therefore, we have added this information in the Discussion section and a timeline table.
|
||
|
Comments 2: The description of Figure 1 (MRI Evolution) could be improved by adding more annotations to each sub-image (1A, 1B, 2A, etc.) to help the reader visually follow the progression. |
||
|
Response 2: Thank you. We have adapted our figure and legend as you have suggested: “Evolution of MRI over time. 1A: T1 image after Gadolinium injection shows a polylobed left frontal brain abscess (48 × 45 mm) with peripheral enhancement on admission before any treatment (surgical or antimicrobial). 1B: T2 image shows a significant perilesional vasogenic oedema at the same moment. 2A and 2B: shows MRI after surgical excision and 6 months of antibiotic treatment with regression of the lesion and of oedema. 3A and 3B: shows stability of the sequelae at 18 months of follow up.”
Comments 3 : There are some minor grammatical errors (e.g., the phrase "ineffective" in line 184 could be replaced with "lack of efficiency"). A simple proofreading by a native English speaker is recommended. Response 3: Agreed. Ongoing proofreading by a native English speaker.
Comments 4 : A concise table summarizing the antimicrobial susceptibility profile (initial E-test versus NRC method) and treatment schedule may help clarify the picture for the reader. Response 4: Agreed. A new table has been added :
Table 1: Comparison of antimicrobial susceptibility results for a Nocardia spp. isolate using gradient strip method (LHUB-ULB) and reference broth microdilution (CNR). Interpretation according to CLSI M62 guidelines. CMI: Minimal Inhibitory Concentration; S: Susceptible; R: Resistant; I: Intermediate.
|
||
|
4. Response to Comments on the Quality of English Language |
||
|
Point 1: There are some minor grammatical errors (e.g., the phrase "ineffective" in line 184 could be replaced with "lack of efficiency"). A simple proofreading by a native English speaker is recommended. |
||
|
Response 1: Agreed. Ongoing proofreading by a native English speaker. |
||
Reviewer 2 Report
Comments and Suggestions for Authors
The paper by E. Lapique et al. on Nocardia brain abscess is a well-written and concise report addressing a rare but clinically significant condition. It is suitable for publication; however, several minor revisions would enhance its overall quality:
-
If available, inclusion of a Gram-stain image of the aspirated fluid would add diagnostic value to the manuscript.
-
The authors should specify the method used for microbial identification at both the genus and species levels, and provide an explanation for the delay in obtaining the final species identification.
-
The statement in the Discussion section — “Management of nocardiosis is essentially based on expert opinion because published literature is limited to case reports, small case series, or in vitro microbiological data.” — requires an appropriate reference.
-
The reference cited in the paragraph: “The gold standard for species-level identification is polymerase chain reaction (PCR) sequencing, specifically targeting the 16S rRNA gene. This method provides reliable species-level identification in the majority of cases (8).” appears to be incorrect and should be verified and corrected.
Author Response
|
Response to Reviewer 2 Comments
|
||
|
1. Summary |
|
|
|
Thank you very much for taking the time to review this manuscript. Please find the detailed responses below and the corresponding corrections highlighted in the re-submitted files.
|
||
|
2. Questions for General Evaluation |
Reviewer’s Evaluation |
Response and Revisions |
|
Does the introduction provide sufficient background and include all relevant references? |
Yes |
|
|
Are all the cited references relevant to the research? |
Yes |
|
|
Is the research design appropriate? |
Yes |
|
|
Are the methods adequately described? |
Could be improved |
Methods have been detailed + new table and image have been added |
|
Are the results clearly presented? |
Yes |
|
|
Are the conclusions supported by the results?
|
Yes |
|
|
3. Point-by-point response to Comments and Suggestions for Authors
|
||
|
Comments 1: If available, inclusion of a Gram-stain image of the aspirated fluid would add diagnostic value to the manuscript. |
||
|
Response 1: Thank you for pointing this out. We agree with this comment. Therefore, we have added this image.
|
||
|
Comments 2: The authors should specify the method used for microbial identification at both the genus and species levels and provide an explanation for the delay in obtaining the final species identification. |
||
|
Response 2: Thank you. We have added this information: “… Nocardia spp. was identified by MALDI-TOF mass spectrometry. However, the initial taxonomic results suggested different species, prompting the need for 16S rRNA gene sequencing to confirm the identification species as Nocardia farcinica.”
Comments 3 : The statement in the Discussion section — “Management of nocardiosis is essentially based on expert opinion because published literature is limited to case reports, small case series, or in vitro microbiological data.” — requires an appropriate reference. Response 3: Agreed. Refences have been added à references 9, 12 and 13. Lebeaux D, Bergeron E, Berthet J, Djadi-Prat J, Mouniée D, Boiron P, et al. Antibiotic susceptibility testing and species identification of Nocardia isolates: a retrospective analysis of data from a French expert laboratory, 2010-2015. Clin Microbiol Infect Off Publ Eur Soc Clin Microbiol Infect Dis. 2019 Apr;25(4):489–95.
Beucler N, Farah K, Choucha A, Meyer M, Fuentes S, Seng P, Dufour H. Nocardia farcinica cerebral abscess: A systematic review of treatment strategies. Neurochirurgie. 2022 Jan;68(1):94-101.
Corsini Campioli C, Castillo Almeida NE, O'Horo JC, Challener D, Go JR, DeSimone DC, Sohail MR. Clinical Presentation, Management, and Outcomes of Patients With Brain Abscess due to Nocardia Species. Open Forum Infect Dis. 2021 Apr 7;8(4):ofab067.
Comments 4 : The reference cited in the paragraph: “The gold standard for species-level identification is polymerase chain reaction (PCR) sequencing, specifically targeting the 16S rRNA gene. This method provides reliable species-level identification in the majority of cases (8).” appears to be incorrect and should be verified and corrected. Response 4: Thank you for pointing this out. The correction has been made à reference 10 : Margalit I, Lebeaux D, Tishler O, Goldberg E, Bishara J, Yahav D, et al. How do I manage nocardiosis? Clin Microbiol Infect. 2021 Apr 1;27(4):550–8.
|
||
Reviewer 3 Report
Comments and Suggestions for Authors
In my view, the introduction is too brief and does not sufficiently familiarize the reader with the topic. The authors should provide a more comprehensive overview of nocardiosis, including a description of its various forms and an outline of the most commonly encountered clinical presentations. It is also important to note that nocardiosis can occur in immunocompetent individuals, a fact that should be acknowledged in the introduction. For reference, see the case report and literature review titled Primary Lymphocutaneous Nocardia brasiliensis in an Immunocompetent Host
Furthermore, how do the authors account for the discrepancy in antimicrobial susceptibility testing (AST) results—specifically, the initial test indicating susceptibility to ceftriaxone and the subsequent test indicating resistance?
The discussion section is excessively long and lacks clarity, making it difficult to follow. I recommend reducing its length by at least 50%, with a focus on whether there is a plausible explanation for the emergence of resistance to ceftriaxone during the course of treatment. Additionally, the patient received imipenem and trimethoprim-sulfamethoxazole (TMP-SMX) for 210 days, which may have been sufficient to manage the infection. Is there any imaging available from the time ceftriaxone therapy was re-initiated?
Lastly, the limitations of the case report and its potential weaknesses have not been adequately addressed by the authors and should be discussed to provide a balanced and transparent analysis.
Author Response
|
Response to Reviewer 3 Comments
|
||
|
1. Summary |
|
|
|
Thank you very much for taking the time to review this manuscript. Please find the detailed responses below and the corresponding corrections highlighted in the re-submitted files.
|
||
|
2. Questions for General Evaluation |
Reviewer’s Evaluation |
Response and Revisions |
|
Does the introduction provide sufficient background and include all relevant references? |
Must be improved |
We have added more information in the introduction |
|
|
|
|
|
Is the research design appropriate? |
Can be improved |
We describe a case report with discrepancies between laboratory and clinical results, no research has been made – could it be possible to be more precise with the remark? thank you |
|
Are the methods adequately described? |
Can be improved |
More details on laboratory methods + a new table have been added |
|
Are the results clearly presented? |
Can be improved |
We have revised presentation + added a new figure + a new table to clarify results |
|
Are the conclusions supported by the results?
|
Must be improved |
We have added new comments to the discussion and conclusion |
|
3. Point-by-point response to Comments and Suggestions for Authors
|
||
|
Comments 1: In my view, the introduction is too brief and does not sufficiently familiarize the reader with the topic. The authors should provide a more comprehensive overview of nocardiosis, including a description of its various forms and an outline of the most commonly encountered clinical presentations. It is also important to note that nocardiosis can occur in immunocompetent individuals, a fact that should be acknowledged in the introduction. For reference, see the case report and literature review titled Primary Lymphocutaneous Nocardia brasiliensis in an Immunocompetent Host |
||
|
Response 1: Thank you for pointing this out. We agree with this comment. More general information has been added in the introduction part.
|
||
|
Comments 2: Furthermore, how do the authors account for the discrepancy in antimicrobial susceptibility testing (AST) results—specifically, the initial test indicating susceptibility to ceftriaxone and the subsequent test indicating resistance? |
||
|
Response 2: Thank you for this comment. Discrepancy is possibly due to different AST methods: we performed gradient strip method while the CNR performed broth microdilution. Both tests were performed on the same initial sample. One explanation for the conflicting ceftriaxone AST results could be related to the production of inducible β-lactamase by Nocardia spp. The expression of this enzyme can vary depending on the experimental conditions.
Comments 3 : The discussion section is excessively long and lacks clarity, making it difficult to follow. I recommend reducing its length by at least 50%, with a focus on whether there is a plausible explanation for the emergence of resistance to ceftriaxone during the course of treatment. Additionally, the patient received imipenem and trimethoprim-sulfamethoxazole (TMP-SMX) for 210 days, which may have been sufficient to manage the infection. Is there any imaging available from the time ceftriaxone therapy was re-initiated? Response 3 : The discussion section has been reduced. As explained in response 2, the discrepancy is not due to the acquisition of resistance, but to different AST methods. The patient only received 10 days of imipenem. We have added a timeline with treatment to be clearer.
Comments 4 : Lastly, the limitations of the case report and its potential weaknesses have not been adequately addressed by the authors and should be discussed to provide a balanced and transparent analysis. Response 4 : We have added a comment in this way on the discussion section.
|
||
Round 2
Reviewer 3 Report
Comments and Suggestions for Authors
I have reviewed the authors’ responses with due diligence. While several of the minor issues raised have been adequately addressed, the central and most critical concern remains unresolved. Regrettably, the authors have not revised the manuscript sufficiently to provide a clear and convincing explanation for the claim made in the title concerning both resistance and therapeutic response. Given that the patient received treatment with multiple antibacterial agents, attributing the curative outcome solely to ceftriaxone appears speculative. This is particularly problematic in light of the documented in vitro resistance, which has not been satisfactorily clarified or justified in the revised version of the manuscript.
Author Response
Thank you for this comment.
We have changed our title for a more nuanced statement.
We are aware that the relationship between the successful response, the received treatment and the AST results are unclear for various reasons already discussed in the manuscript.
However, as clinicians, we think this lack of clarity is precisely the interest of our case report as it reflects real life practice.